# Assessment of Strains Produced by Thermal Expansion in Printed Circuit Boards

**DOI:** 10.3390/ma15113916

**Published:** 2022-05-31

**Authors:** Alexandru Falk, Octavian Pop, Jérôme Dopeux, Liviu Marsavina

**Affiliations:** 1Department of Mechanics and Strength of Materials, University Politehnica Timisoara, 300222 Timisoara, Romania; alexandru.falk@yahoo.com; 2Laboratoire de Génie Civil et Construction Durable (GC2D), University of Limoges, 19300 Egletons, France; ion-octavian.pop@unilim.fr (O.P.); jerome.dopeux@unilim.fr (J.D.)

**Keywords:** DIC, PCB, principal strain, thermal expansion, strain gauge rosette

## Abstract

The paper proposed an alternative optical metrology to classical methods (strain gauge measurements and numerical simulation) for strain determination on printed circuit board (PCBs) due to thermal loads. The digital image correlation (DIC) technique was employed to record the strain distribution in some particular areas of the PCB. A thermal load was applied using a heating chamber, and the measurements were performed at four different temperature steps (25 °C, 50 °C, 85 °C and 120 °C). An increase in the principal strains with temperature was observed. For validation, the principal strains on the PCB obtained with DIC were compared with the values from gauge strain measurements and numerical simulation. The conclusions highlighted that DIC represents a technique with potential for strain measurement caused by thermal deformation, with the advantages of full field measurement, less preparation of the surface and good accuracy.

## 1. Introduction

A printed circuit board (PCB) is the board base for physically supporting and wiring the surface-mounted and socketed components in most electronics. Most PCBs are made from fiberglass or glass-reinforced plastics with copper traces.

The main causes that induce strains on PCB are the surfaces on which the PCB is placed, which can be at different levels, the assembly process of electronic components, impacts, vibrations and temperature variation. All of this can lead to failures in microprocessor ball grid arrays (BGAs) and route damage of the electronic components [1,2,3].

Thermal stresses are induced due to mismatch of the coefficients of thermal expansion (CTE) of component materials during temperature variations. Other sources of thermal stresses can be the non-uniform temperature distribution in the components and the anisotropy of thermal expansion in composite materials. The thermo-mechanical deformations produced by thermal expansion represent one of the most important roots of failure in electronic components. The mismatch of the CTE also can produce package-related failures such as die cracking, bond fractures and lift-off [4]. The understanding of thermo-mechanical induced deformations on the PCB is important in order to predict the reliability of electronic assembly.

For strain measurement on PCBs, the current method used in industry is the resistive strain gauge method presented in [5,6] and, according to documentation, has been used for strain measurement in many applications [7,8,9]. Using this method, the value of strain in known only at the points where the strain gauges are located and the determination of principal strains requires the use of strain gauge rosettes. In order to have full field strain measurement, several optical techniques were developed, such as the digital image correlation, mark tracking or grid methods, which are better alternatives for characterizing mechanical behavior in the case of PCBs [10,11,12].

According to documentation, digital image correlation (DIC) was successfully employed to determine the thermal deformations in many applications [13,14,15,16,17,18,19]. The DIC technique also was used for strain measurement on PCBs [20,21,22] and other composite materials [23,24,25,26,27].

The scope of this paper was to investigate if the DIC technique can be used for full field measurement of the strains caused by thermal expansion in PCBs and to validate the results comparing with the results from finite element analysis (FEA) and strain gauge measurements.

## 2. Materials and Methods

### 2.1. Digital Image Correlation (DIC)

In the digital image correlation (DIC) technique, the strains are obtained directly from acquired images based on a correlation algorithm. Displacements are determined from a set of images on the object surface taken before and after deformation using digital cameras by searching the position of a subset in an image after deformation. This principle is based on the assumptions that the features of an object surface are displaced together with the object surface and that they are preserved after deformation.

Displacements are determined by looking for an area for which the gray distribution is the same as the gray distribution of the subset before deformation. For this reason, an object surface must have a random pattern. Figure 1 shows an arbitrary pattern of an object surface before and after deformation. Non-uniform gray levels are distributed in a subset extracted from an image before deformation because a random pattern on an object surface is recorded. An area with gray levels that are the same as those of the subset is sought in the image after deformation. Then, the subset position after deformation is found.

Regarding the paint used, it was chosen to withstand temperatures above 120 °C, in order to guarantee quality of the black and white speckle pattern during the thermal expansion of the PCB. As mentioned before, the PCBs were subjected to thermal stresses of 120 °C.

The optical measurements were realized using an optical device, which comprised a charge-coupled device (CCD) camera (resolution: 3840 × 2748) with a Pentax zoom lens from 12.5 at 75 mm. The signal-to-noise ratio of the charge-coupled device camera was about 45.21 dB.

### 2.2. Experimental Setup

The PCBs represent composite structures made of FR-4 composite plate, solder mask, and copper. Figure 2 presents a double-sided PCB made of two copper layers mounted on FR4 substrate (where FR stands for flame retardant and the number ‘4’ indicates woven glass-reinforced epoxy resin) and covered with solder mask and other layers.

The investigated PCBs were assembled between housing and cover with four M2.5 screws. According to Figure 2, bumps that are 0.2 mm higher than the PCB seating surface have been provided on the surfaces of the housing and the cover; these lead to the bending of the PCB, which plays an important role in the deformation state of the PCB. The screw driving sequence for the four screws was 0.25 Nm, 0.4 Nm, 0.55 Nm, and 0.7 Nm, respectively, applied progressively using an electronic screwdriver (see Figure 2). So, the DIC and strain gauge measurements were performed when all four screws were torqued to 0.7 Nm.

The presence of bumps and their different heights produced bending of the PCB. The real geometry, boundary conditions, loading scenario for the screw driving sequence and temperature variation were modeled in the finite element analysis.

In order to observe the evolution of strain as a function of temperature, the PCB was placed in an oven, and the temperature was increased in several steps: 25, 50, 85 and 120 °C.

As part of this study, we proposed a complementary analysis of the mechanical behavior of PCBs subjected to mixed mechanical and thermal stress, using several measurement and numerical modeling methods. Two experimental methods were employed to measure the strains: strain gauge rosettes and digital image correlation. The finite element method was employed for the numerical analysis.

The experimental set-up for 2D digital image correlation and strain gauge measurements is presented in Figure 3, including the sample, climatic room (Heratherm OGS180 with gravity convection and 50–250 °C temperature range) and the CCD camera. The illumination system allowed us to obtain a homogenous intensity distribution in the ROI without heating the sample, which ensured excellent measurement conditions. The software used for image acquisition was Trasse ANDRA3, and for digital image correlation, Correla, developed by the University of Poitiers. As explain below, the DIC was chosen to measure the PCB local and global deformation during the thermal loading. The displacement and strain fields were measured during and at the end of each thermal sequence.

In addition, with the optical device, the strain gauges were mounted on the PCB. As illustrated in Figure 3, the experimental setup for strain gauge measurements consisted of the PCB specimen, two strain gauge rosettes mounted near the big component, and the data acquisition system. The connection used was a quarter bridge in Spider 8 data acquisition system. Considering the thermal expansion of the PCB during thermal loading, the temperature compensation in the strain gauge was operated.

In order to compare the optical metrologies with the strain gauges’ measurements, both were synchronized.

## 3. Results

### 3.1. DIC Measurements

According to Figure 4, a region of interest (ROI), marked with blue, in which the strain was obtained and two paths (optical gauges), marked with red, in the area of the microprocessor corners, was considered, with A1–A2 and B1–B2 in the same locations where the strain gauges were placed, and where the results were analyzed. Correla software was used for correlation and analysis of the acquired images.

The strain fields measured for all thermal loadings, obtained in the region of interest (ROI), are displayed in Figure 5.

The analyses of the principal strain maps plotted in Figure 5 highlighted the PCB heterogeneity, amplified by the presence of electronic corposants. It should be observed that for the temperature exceeding 85 °C, the strain reached 700 microstrains.

In order to compare the DIC measurements with those of the strain gauges (see Figure 3), two paths, A1–A2 and B1–B2, were defined, as illustrated in Figure 4. Figure 6 shows the principal strain evolution along the paths A1–A2 and B1–B2 in accordance with the temperature evolution.

The principal strain is evaluated as:
(1)
$$\varepsilon_{max}=\frac{\varepsilon_{x}+\varepsilon_{y}}{2}+{\left[{\left(\frac{\varepsilon_{x}+\varepsilon_{y}}{2}\right)}^{2}+{\left(\frac{\gamma_{xy}}{2}\right)}^{2}\right]}^{0,5}$$

where 
$\varepsilon_{x}$
, 
$\varepsilon_{y}$
 are the strains measured in the x and y directions and 
$\gamma_{xy}\;$
 is the shear strain.

As in the precedent case (Figure 5), an increase could be observed in the strain with each step of temperature increase. Increasing the temperature to 85 °C induced a maximum principal strain above 900 microstrains, which was higher than the allowable value of 700 microstrains [5].

The strain evolution along the A1–A2 and B1–B2 paths revealed the influence of electronic components positioned near both paths. As can be observed, the mechanical behavior of the PCB along the A1–A2 and B1–B2 paths was not homogeneous. The strain distribution also showed the orthotropic properties of the FR4 substrate. This behavior was more evident at the higher temperatures. This heterogeneous deformation of the PCB, as well as its mounting on the housing with four screws, could generate torsion of the plate, resulting in the detachment of electronic components.

### 3.2. Strain Gauge Measurement

As indicated in the introduction, in addition to DIC, the PCB deformation under thermal solicitation was also investigated using strain gauge rosettes.

The strain gauge rosettes were positioned at the microprocessor (biggest component) corners of the PCB, as illustrated in Figure 3. The strain gauges used were Kyowa KFGS-1-120-D17-11 models (right-angled gauge rosette) with the following characteristics: 120 Ω resistance, three gauges placed at 0°, 45°, and 90° angles. Strains were recorded using a Spider 8 data acquisition system and analyzed with Catman Easy V5.3.1 software.

The principal strain was obtained with Equation (2) using the strain measurements from the gauges positioned at 0°, 45°, 90° angles and plotted in Figure 7.

(2)
$$\varepsilon_{1,2}=\frac{\varepsilon_{0}+\varepsilon_{45}}{2}\pm \frac{1}{2}\sqrt{{\left(\varepsilon_{0}-\varepsilon_{45}\right)}^{2}+{\left(\varepsilon_{45}-\varepsilon_{90}\right)}^{2}}$$

where *ε*_0, 45 and 90_ are the deformations corresponding to the gauges positioned at 0°, 45°, 90° angles.

Based on the strain gauge rosettes and Equation (2), Figure 7 presents the variation in principal strain measured during temperature evolution.

As in the case of DIC, we could observe a difference in the mechanical behavior measured by both gauges. The differences in strain measured in the two areas were lower, having a maximum difference of 20.6% at 120 °C. As explained above, this difference could be caused by the PCB heterogeneity, due to a local stiffening caused by the presence of electronic components. The difference between the coefficients of thermal expansion of FR4 and electronic component materials could also explain this difference in mechanical behavior. The DIC measurements plotted in Figure 6 also revealed a difference between both investigation paths.

In Figure 7, we can also observe the PCB thermal stabilization corresponding to each thermal sequence. It can be observed that at the end of each temperature level, the deformation no longer evolves.

The comparison of DIC and strain gauge measurements and of the maximum value of maximum principal strain are plotted in Figure 8.

The results reveal that for region 0 (A1–A2 and strain gauge 0), the maximum difference of 30.4% occurred at 50 °C; for region 1 (path B1–B2 and strain gauge 1) the maximum difference of 12.5% was also at 50 °C. This difference could be explaining by the presence of experimental noises in the case of DIC measurements. The light source, the environmental conditions (reflections, heat vapors, air temperature), but most importantly, the climatic room vibrations could sometimes affect the accuracy of DIC.

Nevertheless, the strain analyses using DIC and strain gauges revealed a good correlation between both approaches. However, the DIC analysis allowed a multiscale analysis of mechanical fields.

### 3.3. FEA Analyses

In addition to the experimental tests, a finite element analysis was performed in order to compare the efficiency of DIC measurements.

The commercial software used for finite element analysis was Ansys Workbench 18.1. According to the documentation, the FEA method is used to determine the strain on electronic components in many applications [28,29,30,31]. FEA allows one to obtain the distribution of strain across the PCB surface.

To reduce the computational time of numerical analysis, the electronic components were defined as simple blocks made of hard plastic material, while the PCB was considered an elastic orthotropic FR4 material. The physical and elastic properties of the materials are presented in Table 1, Table 2 and Table 3, as provided by the manufacturer at room temperature (23 °C).

The boundary conditions were applied to the screws as bolt pretension of 1800 N equivalent of 0.7 Nm; this value was chosen according to the screw supplier based on their simulations. As in the experimental case, four temperature steps of 25, 50, 85 and 120 °C were also imposed. A steady-state type of thermal analysis was performed.

Figure 9 shows the meshing of the PCB, consisting of 80,254 tetrahedral elements, connected in 266,649 nodes.

The principal strain on PCB was obtained after numerical analysis. The results for the four considered temperatures are shown in Figure 10.

The strain maps highlight an increase in strains around the microprocessor (the biggest component on the PCB). The numerical analysis showed that the maximum allowable limit of 700 μ strains [6] was exceeded in both areas (0 and 1) at 85 °C.

Now, if we compare the strain maps obtained by DIC and the finite element method, we can observe a similitude in the strain distribution and amplitude, as illustrated in Figure 11.

As in the case of the DIC and strain gauge rosette investigations, the results from FEA were analyzed in the area of the microprocessor corners according to the paths A1–A2 and B1–B2 (see Figure 12).

The values of principal strains for the considered temperatures are shown in Figure 13.

The numerical results showed an increase in the principal strain as the temperature increased. In the case of the A1–A2 path, a linear increase could be observed with distance at temperatures higher than 50 °C. A more constant principal strain was observed on the B1–B2 path at all considered temperature levels.

## 4. Discussion

According to Figure 6, Figure 7 and Figure 13, the principal strains measured with the DIC technique were in the same range as the strain gauge measurements and finite element simulations.

For a better understanding, Figure 14 shows a comparison of principal strain obtained from DIC measurements and FEA analyses on the paths A1–A2 and B1–B2, whereas Figure 8 shows a comparison between the maximum values of maximum principal strain obtained from DIC measurements and strain gauge measurements. Relatively good agreement was obtained between the results from DIC measurements and FEA simulations. For the A1–A2 path, slightly larger differences could be observed at 50 °C, where the maximum difference was 16.8%, and at 120 °C was 14.1%. For the B1–B2 path, slightly larger differences could be observed at 120 °C, where the maximum difference was 22.6%.

The prescribed allowable strain limit on PCB of 700 microstrains [6] was reached starting from the 85 °C temperature.

The DIC measurements could be successfully used to validate the FEA results obtained in the design stage of PCBs. Additionally, the DIC technique could replace the actual strain gauge measurements, taking advantage of less surface preparation and full field strain results.

The strain measurement methodology based on DIC appears more reliable and accurate for evaluation of the distribution of strains and their monitoring during testing and qualification.

## 5. Conclusions

The present study revealed the influence of temperature on the mechanical behavior of printed circuit boards (PCBs). The numerical and experimental data obtained for four different temperatures demonstrated the sensitivity of PCBs to thermal solicitation and strain distribution.

The strain analysis was achieved using several approaches and methods. For the experimental case, the strain evolution was monitored using digital image correlation and strain gauges. The experimental results revealed good agreement between both experimental techniques. However, the DIC analysis allowed a multiscale analysis of strain distribution during the PCB thermal loading. The strain distribution obtained by DIC also showed a strain intensity in the vicinity of electronic components bonded on the printed circuit boards.

This experimental analysis was completed with numerical simulations based on the finite element method. The finite element mesh corresponded to the PCB geometry and configuration, and the boundary conditions were the same as those in the experimental case. The numerical results showed the same tendency as the experimental measurements.

The strain analysis showed that starting from 85 °C, the strain level reached 700 microstrains and that the printed circuit boards began to incur damage. It should be noted that beyond this value, there was a high risk of delamination of the electronic components.

The use of DIC for full-field analysis of strains in PCBs could be adopted at industrial scale to measure and monitor the strain fields.

## Figures and Tables

**Figure 1 materials-15-03916-f001:**
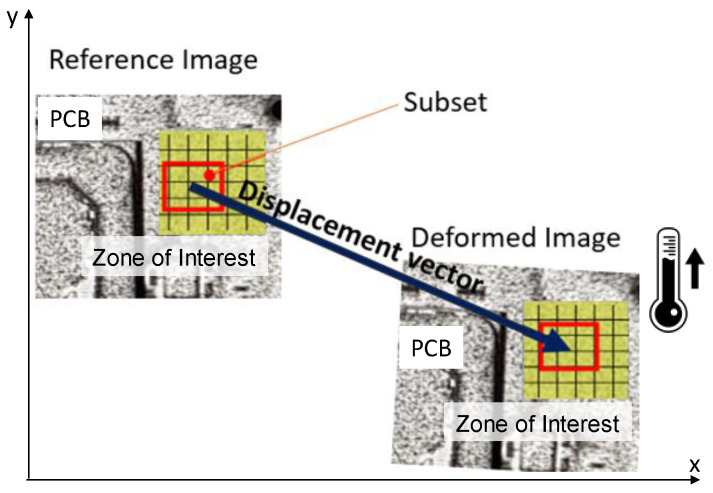
The principle of the DIC technique.

**Figure 2 materials-15-03916-f002:**
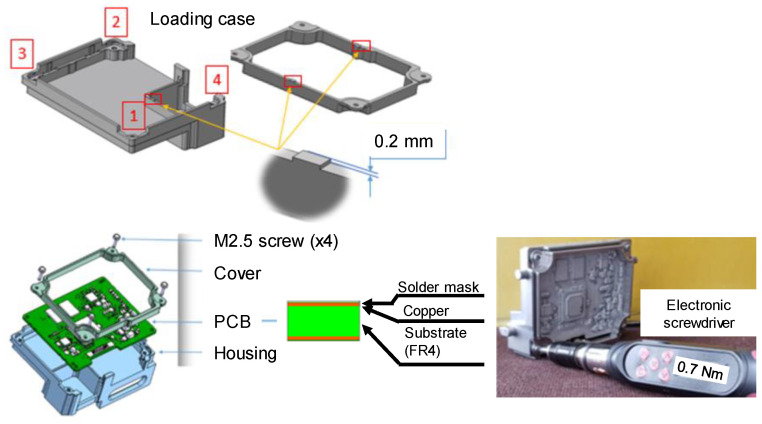
The PCB assembly between housing and cover. The electronic screwdriver for tightening the assembly.

**Figure 3 materials-15-03916-f003:**
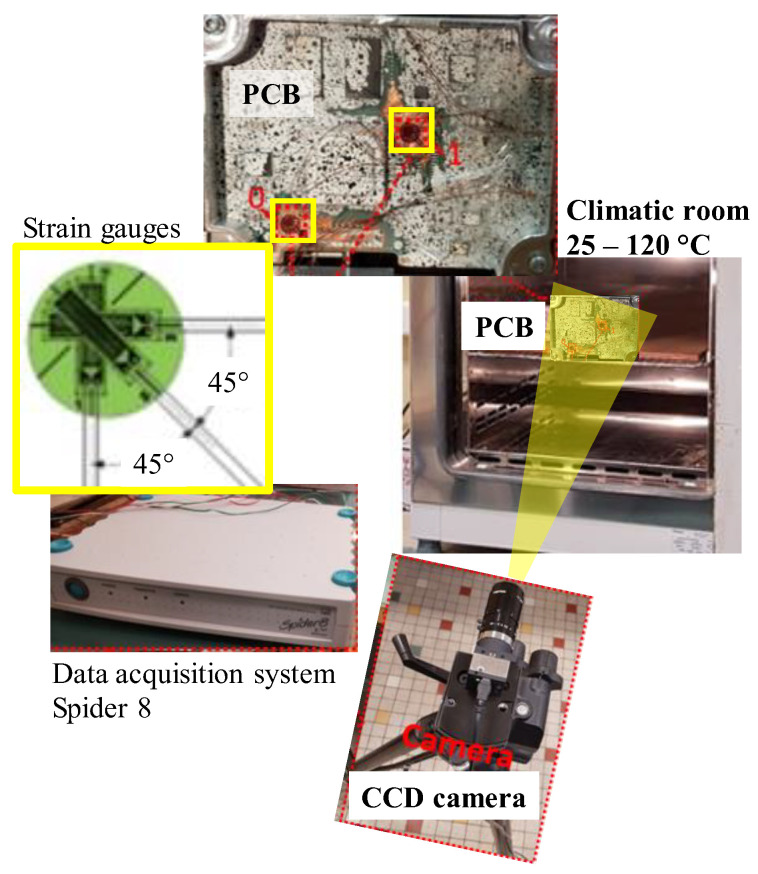
The experimental setup for DIC measurements.

**Figure 4 materials-15-03916-f004:**
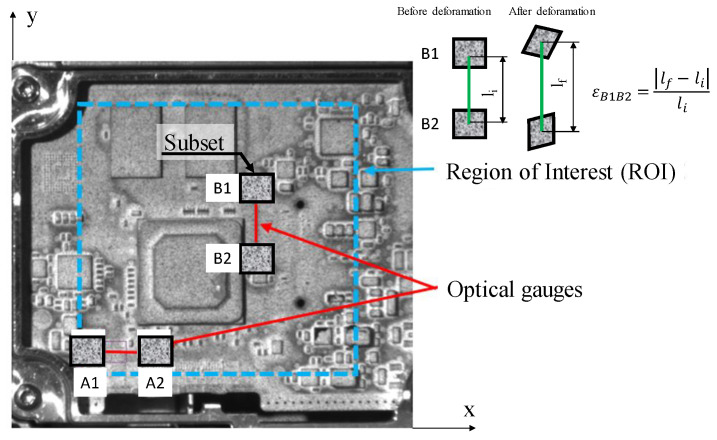
DIC measurement areas.

**Figure 5 materials-15-03916-f005:**
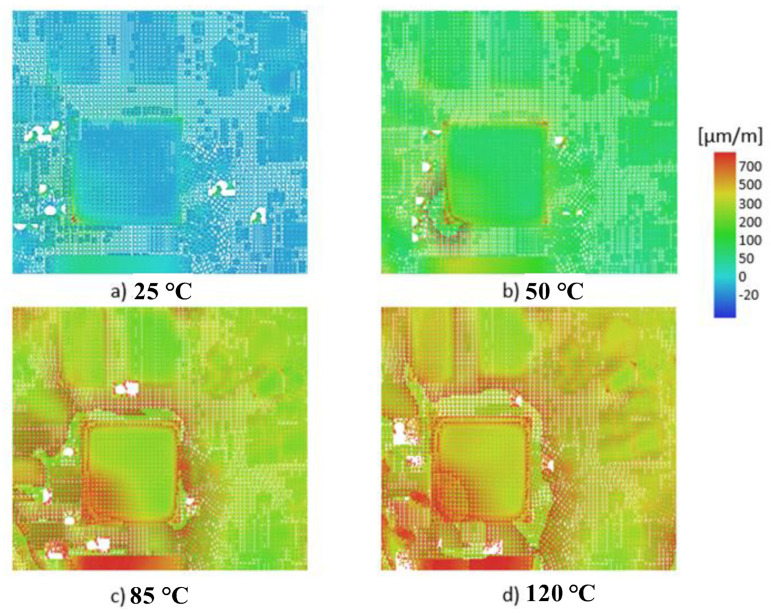
The principal strain DIC in ROI: (**a**) 25 °C; (**b**) 50 °C; (**c**) 85 °C; (**d**) 120 °C.

**Figure 6 materials-15-03916-f006:**
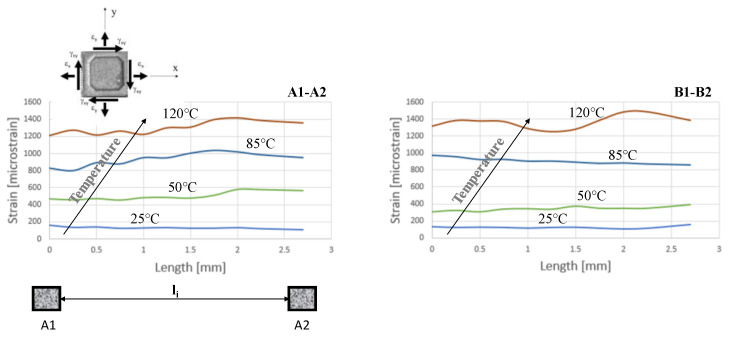
Principal strain DIC according to A1–A2 and B1–B2 paths.

**Figure 7 materials-15-03916-f007:**
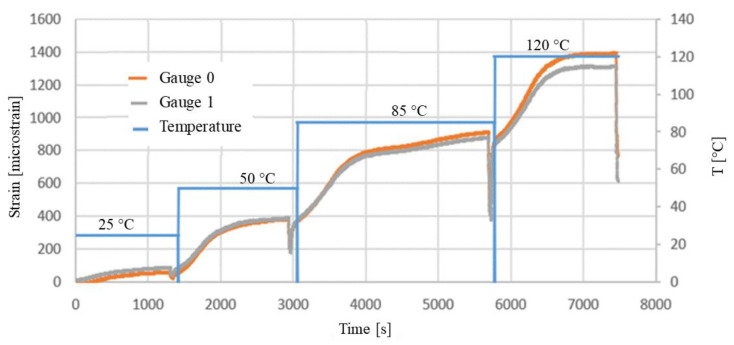
The principal strain-strain gauge measurement.

**Figure 8 materials-15-03916-f008:**
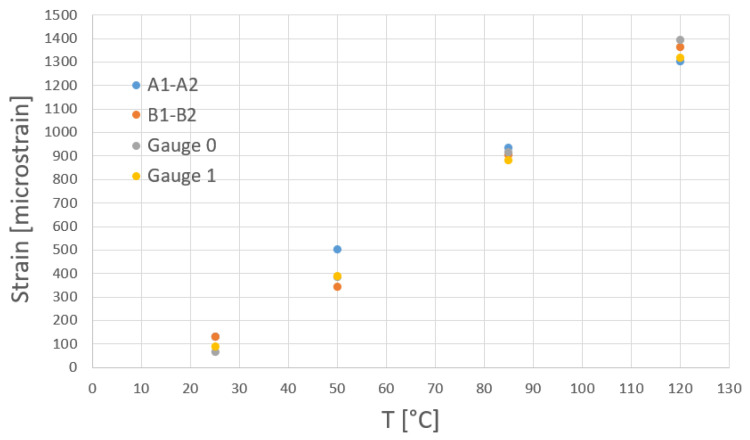
Comparison between maximum values of maximum principal strain: DIC vs. strain gauge measurement.

**Figure 9 materials-15-03916-f009:**
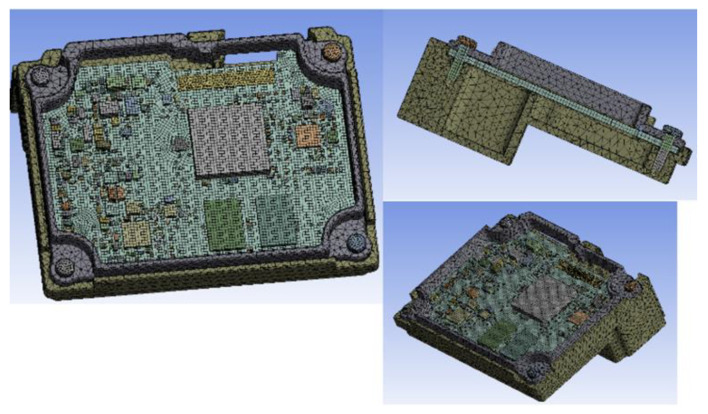
FEA mesh.

**Figure 10 materials-15-03916-f010:**
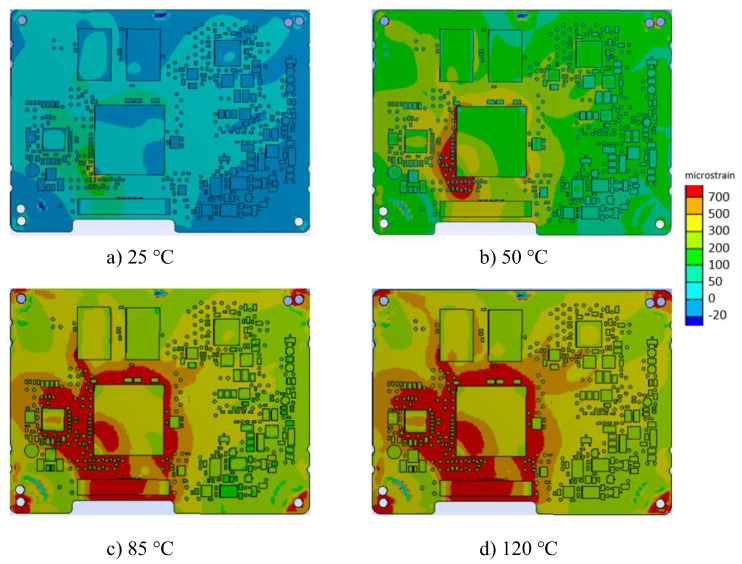
Principal strain obtained by finite element method: (**a**) 25 °C; (**b**) 50 °C; (**c**) 85 °C; (**d**) 120 °C.

**Figure 11 materials-15-03916-f011:**
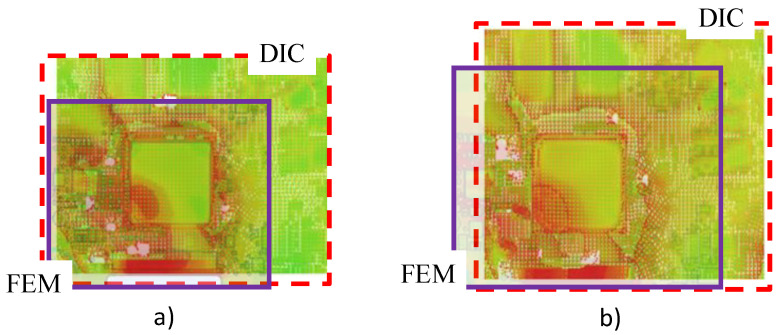
Comparison of DIC and FEM for: (**a**) 85 °C; (**b**) 120 °C.

**Figure 12 materials-15-03916-f012:**
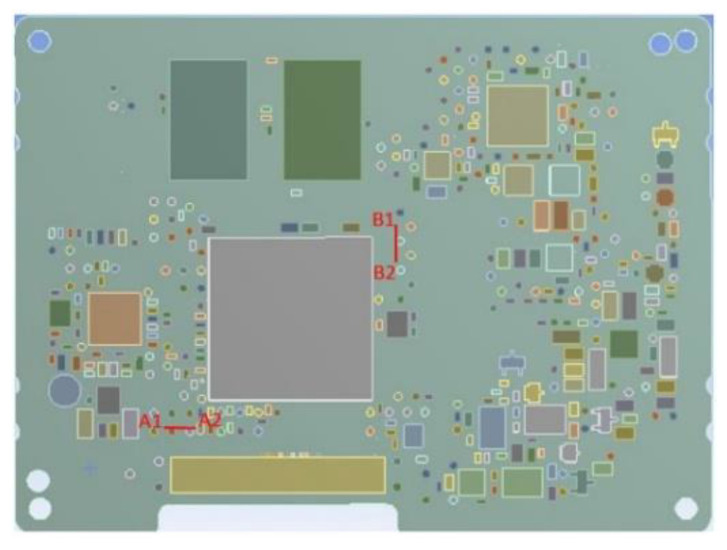
Definition of strain evaluation zones.

**Figure 13 materials-15-03916-f013:**
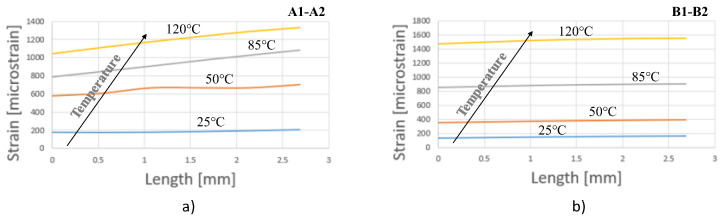
The variation in strain obtained by FEA vs. temperature: (**a**) A1–A2; (**b**) B1–B2.

**Figure 14 materials-15-03916-f014:**
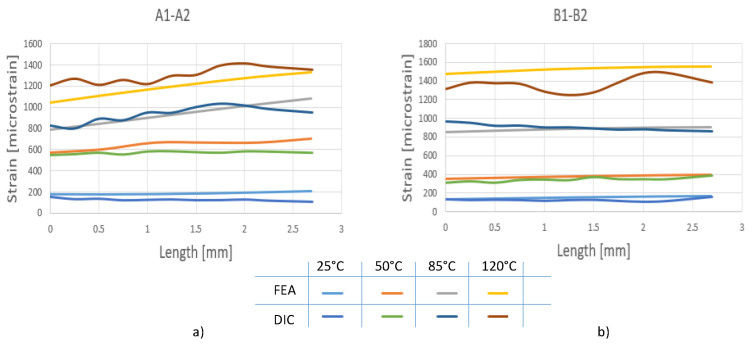
DIC vs. FEA—Comparison of principal strain: (**a**) A1–A2 path; (**b**) B1–B2 path.

**Table 1 materials-15-03916-t001:** The physical and elastic properties of FR4 material.

Property	Symbol	Unit	Value
Density	ρ	g/cm^3^	1.85
Orthotropic Instantaneous Coefficient of Thermal Expansion	α_x_ α_y_ α_z_	°C^−1^ °C^−1^ °C^−1^	1.35 × 10^−^^5^ 1.35 × 10^−^^5^ 4.50 × 10^−^^5^
Longitudinal Modulus of Elasticity	E_X_ E_y_ E_z_	MPa MPa MPa	1.69 × 10^4^ 1.69 × 10^4^ 7.40 × 10^4^
Poisson’s Ratio	ν_xy_	-	0.11
	ν_yz_	-	0.39
	ν_zx_	-	0.39
Shear Modulus	G_xy_	MPa	7.60 × 10^3^
	G_yz_	MPa	3.30 × 10^3^
	G_zx_	MPa	3.30 × 10^3^

**Table 2 materials-15-03916-t002:** The physical and elastic properties of electronic component material.

Property	Symbol	Unit	Value
Density	ρ	g/cm^3^	1.63
Coefficient of Thermal Expansion	α	°C^−1^	6 × 10^−^^5^
Young’s Modulus	E	MPa	2.55 × 10^4^
Poisson’s Ratio	ν		1.10 × 10^−^^1^
Bulk Modulus	B	MPa	1.70 × 10^4^
Shear Modulus	G	MPa	1.02 × 10^4^

**Table 3 materials-15-03916-t003:** The physical and elastic properties of housing and cover material.

Property	Symbol	Unit	Value
Coefficient of Thermal Expansion	α	°C^−1^	2.30 × 10^−^^5^
Density	ρ	g/cm^3^	2.7
Young’s Modulus	E	MPa	2.55 × 10^4^
Poisson’s Ratio	ν		1.10 × 10^−1^
Bulk Modulus	B	MPa	1.70 × 10^4^
Shear Modulus	G	MPa	1.02 × 10^4^

## Data Availability

Not applicable.

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
