# Peer review of "Assessment of Strains Produced by Thermal Expansion in Printed Circuit Boards"

_materials, 2022, doi:10.3390/ma15113916_

Round 1

Reviewer 1 Report

Review Report on Manuscript ID: materials-1713100

Recommendation: major revision

Type: Article

Title:

 The assessment of the strains produced by thermal expansion in Printed Circuit Boards

Authors:  Alexandru Falk, Octavian Pop, Jérôme Dopeux

Corresponding Auhor:  Liviu Marsavina

 General statement

The paper a continuation of the authors' works [20,21]. This time, the aim of the work was to use the 2D Digital Image Correlation (DIC) technique to measure deformation in the full cross-sectional area caused by thermal expansion in the range 25-120 degC of printed circuit boards and to validate the DIC results in comparison with the results of the finite element analysis (FEM) and strain gauge measurements. The results of the work are interesting but there is no displacement measurement errors analysis and sensitivity analysis of the results obtained by the FEM method.  In my opinion the present form of the manuscript needs major revision which results from the comments below.

Remarks

  1. Lines 171-173 the authors write ‘ For region 0 (A1-A2 and strain gauge 0) the maximum difference of 30.4% is at 50 0C respectively for region 1 (path B1-B2 and strain gauge 1) the maximum difference of 12.5% is at 50 0C.’ and further (Line 178-179) ‘ The strain analysis operated using DIC and strain gauges reveals a good correlation between both approaches’ -  what factors could cause the differences of maximum principal strains and can the 30.4% maximum difference for region 0 be considered as good correlation between DIC and strain gauge approaches?
  2. What is the impact of speckle pattern, non-parallelism between the CCD target and the object surface and out-of-plane displacement on determined by 2D DIC technique strains of  PCB?
  3. Almost nothing is known about the climatic room shown in Fig. 3 (including the name of the manufacturer, accuracy of temperature stabilization, air flow used in the test, how long the temperature was stabilized)
  4. What is the signal-to-noise ratio of the used CCD camera
  5. How the measurement time was selected to determine the maximum principal strain during strain gauge measurement (Fig. 7)
  6. At what temperature the data in tables 1-3 are given?
  7. FEA analyses were not carried out in terms of determining the sensitivity of the solution to input data disturbances. For this reason, it is difficult to refer to the results given in Fig. 13

Detailed comments

  • Line 79 , copper instead of Copper
  • Line 123, the results were analysed instead of  ‘the results were interrogated’
  • Line 128, Figure 6 is displayed the investigation instead of ‘Figure 6 is displayed the interrogation’
  • Line 140, invalid symbol for Celsius scale
  • Tables 1-3, invalid symbol for Celsius scale
  • Line 210, Results were analysed  instead of ‘Results were interrogated’
  • Line 229, Figure 14 is missing
  • Lines 232-234,  invalid symbol for Celsius scale

Reviewer 2 Report

1 Please revise those numbers in tables by replacing decimal with multiplication sign.

2 Please rewrite your abbreviations more carefully with a unified form.

3 Please emphasize the novelty of this manuscript as DIC and strain gauge methods are very basic methods which has been adopted widely. Frankly, I cannot catch the novelty of this manuscript.

4 The quality of this manuscript needs significantly improved as there are so many format errors.

5 It is more meaningful if the investigation is focused on stress or strain intensity of vincinity for PCB.

Round 2

Reviewer 1 Report

I found the authors' responses to my comments to be satisfactory. I appreciate the effort of the authors demonstrated in the revised version of the article and wish them success in the scientific field.

Reviewer 2 Report

No further comments